# Acceptability and Feasibility of Maternal Mental Health Assessment When Managing Small, Nutritionally At-Risk Infants Aged < 6 Months: A Key Informant Interview Study [note 1]

**DOI:** 10.3390/children11020209

**Published:** 2024-02-06

**Authors:** Natalie Mee, Mubarek Abera, Marko Kerac

**Affiliations:** 1Department of Population Health, Faculty of Epidemiology and Population Health, London School of Hygiene and Tropical Medicine, Keppel Street, London WC1E 7HT, UK; natalie.mee1@alumni.lshtm.ac.uk; 2Department of Psychiatry, Faculty of Medical Sciences, Institute of Health, Jimma University, Jimma P.O. Box 378, Ethiopia; mubarek.abera@ju.edu.et

**Keywords:** mental health, screening assessments, maternal health, infant health, nutritionally at-risk, nutrition programmes, low–middle income countries, MAMI care pathway

## Abstract

Maternal mental health (MMH) conditions and infant malnutrition are both major global public health concerns. Despite a well-established link between the two, many nutrition programmes do not routinely consider MMH. New World Health Organization (WHO) malnutrition guidelines do, however, emphasise MMH. To inform guideline rollout, we aimed to assess the feasibility and acceptability of MMH assessments in nutrition programmes in low-resource settings. Ten semi-structured interviews were conducted with international key informants who work on nutrition programmes or MMH research. Interview transcripts were coded using subthemes derived from the key points discussed. The benefits and risks were highlighted. These included ethical dilemmas of asking about MMH if local treatment services are suboptimal. Commonly reported challenges included governance, staff training and finance. Community and programme staff perceptions of MMH were primarily negative across the different settings. Many points were raised for improvements and innovations in practice, but fundamental developments were related to governance, care pathways, advocacy, training, funding and using existing community networks. Future implementation research is needed to understand whether assessment is safe/beneficial (as it is in other settings) to promote MMH screening. Current service providers in low-resource settings can undertake several steps, as recommended in this paper, to improve the care offered to mothers and infants.

## 1. Introduction

Maternal mental health (MMH) problems and child malnutrition are both major public health problems affecting millions worldwide [1]. MMH conditions are among the commonest complications of a pregnancy and can arise de novo, as well as affecting those with prior risk and prior episodes of illness [2]. Depression is the most prevalent MMH concern and, according to a 2023 systematic review of the global literature, affects 26% of mothers in the perinatal period, 29% antenatally and 28% in postnatally [3]. 

Child malnutrition is also a global concern. Latest WHO/UNICEF/World Bank figures estimate that 22% of children aged under 5 years (child u5) worldwide are stunted (i.e., too short for their age, a marker of chronic malnutrition), and 7% are wasted (i.e., too thin for their height, commonly seen as a marker of acute malnutrition [4]. Infants aged under 6 months (infants u6m) are particularly at risk, with some 18% stunted and 21% wasted [5]. 

Despite a solid and growing evidence base linking MMH with infant/child malnutrition [6,7,8,9], in many settings the two conditions are treated separately in different programmes, run by different health professionals with different skill sets and who specialise in one or the other [10,11]. This can result in poor and inconsistent care and missed opportunities, which can be devastating on many fronts. The major immediate risk of malnutrition is mortality: malnutrition accounts for some 45% of all deaths among children u5 worldwide [12]. There are also longer-term risks, including poor cognitive and behavioural development [13], and even the risk of cardiometabolic noncommunicable disease in much later adult life [14]. For mothers, undiagnosed or poorly treated MMH can affect their physical health; impact on their relationships with others; increase risky behaviours; and even, in some cases, lead to death [15]. MMH problems also impair mother–child interactions and caring practices and independently affect infant health, sleep and development [15]. One study also shows a link with infant temperament [16]. Other adverse effects of poor MMH are on breastfeeding [17,18,19]; this is one of the main mechanisms explaining the subsequent impact on infant malnutrition. There is an important and urgent need to overcome the MMH/malnutrition divide.

New 2023 WHO guidelines on the “Prevention and management of wasting and nutritional oedema (acute malnutrition) in infants and children” are notable for the attempts to better integrate and link MMH and malnutrition issues [20]. Though details are limited, MMH features strongly throughout the guidelines and notably so in the chapter on infants u6m. For example (italics added for emphasis):“Mothers/caregivers and their infants less than 6 months of age at risk of poor growth and development should receive regular care and monitoring by health professionals. The immediate goal is the early detection of any acute medical *or psychological problems*… The longer-term goal of this regular care and monitoring is to enable these infants to grow and develop …*whilst simultaneously supporting their mothers/caregivers* with their own health and *wellbeing*. This approach recognizes the importance of acknowledging and *caring for the mother/caingregiver and infant as an inter-dependent pair for both to survive and thrive*” (Good Practice Statement A1);“Decisions about whether an infant less than 6 months of age at risk of poor growth and development needs a supplementary milk in addition to breastfeeding must be based on a comprehensive assessment of the medical and nutritional/feeding needs of the infant, *as well as the* physical and *mental health of the mother/caregiver*” (Good Practice Statement A6);“A*ssessment* of the physical and *mental health status of mothers or caregivers should be promoted and relevant treatment or support provided*” (Recommendation A7, covering infants admitted to inpatient care);“Among mothers/caregivers of infants less than 6 months of age at risk of poor growth and development, comprehensive assessment and support are recommended to *ensure maternal/caregiver* physical and *mental health and wellbeing*. These actions are also important to optimize growth and development in infants at risk of poor growth and development” (Good Practice Statement A8).

MMH also features in the case definition of small and nutritionally at-risk infants u6m (referred to by the WHO as “infants at risk of poor growth and development”), determining who should be admitted to a treatment/prevention programme: “Infants with known risk factors for poor growth and development: *Maternal risk—*physical or *mental health problem(s)* affecting caring practices” [20]. 

Whilst the new WHO malnutrition guidelines offer an important international policy-level push for better future MMH/nutrition service linkages, detailed descriptions of how such integration might work in front-line clinical services are currently lacking. Governments, NGOs (nongovernmental organisations) and others implementing new guidelines in their malnutrition treatment and prevention programmes are faced with numerous practical and ethical challenges. One such challenge is how much to even ask patients about MMH if local MMH treatment services are limited, as is the case in many low- and middle-income countries (LMICs) [21]. Some argue that even the small step of asking about MMH is significant: it helps a mother know that her feelings matter; it helps highlight the issue of MMH; it provides local data and local experience. For example, MMH assessment features strongly in the “MAMI Clinical Care Pathway” (see Figure 1), which aims to improve the management of small and nutritionally at-risk infants under 6 months and their mothers [22,23]. MAMI offers basic guidance for how to both identify and treat common MMH conditions. The rationale is that highlighting MMH at this early stage will not only benefit patients directly but will ultimately end in improved MMH services once the need is more clearly seen and documented. Others take opposite perspectives: MMH services must come first; flagging a problem in the absence of an appropriate treatment is unethical. There is, thus, a difficult balance between screening MMH, offering suboptimal support, not asking about mental health and MMH conditions remaining unaddressed. 

### Study Rationale and Aims

As governments, NGOs and others move forward to operationalise and roll-out new WHO malnutrition guidelines, they need to consider how to incorporate the elements of MMH assessment and treatment. Evidence on the likely acceptability and feasibility of this is currently lacking.

This study aimed to fill this evidence gap by determining the feasibility, acceptability and perceived benefits and risks of asking about MMH as part of a routine assessment for infant malnutrition, especially in settings where specialist MMH services are unavailable, limited or difficult to access. 

Towards this, the specific objectives were: To ascertain whether assessing MMH is regarded as acceptable by communities, mothers and staff working in different settings (specifically, is asking about suicidal intent acceptable and appropriate?).To highlight any challenges to the feasibility of implementing MMH assessments.To identify the current perceived risks of assessing MMH, including the risks to mothers, infants and patient-facing staff.To identify the perceived benefits of assessing MMH, including the benefits to mothers, infants and patient-facing staff.To identify the tangible next steps needed to improve the feasibility, acceptability and accuracy of mental health assessments.

**Figure 1 children-11-00209-f001:**
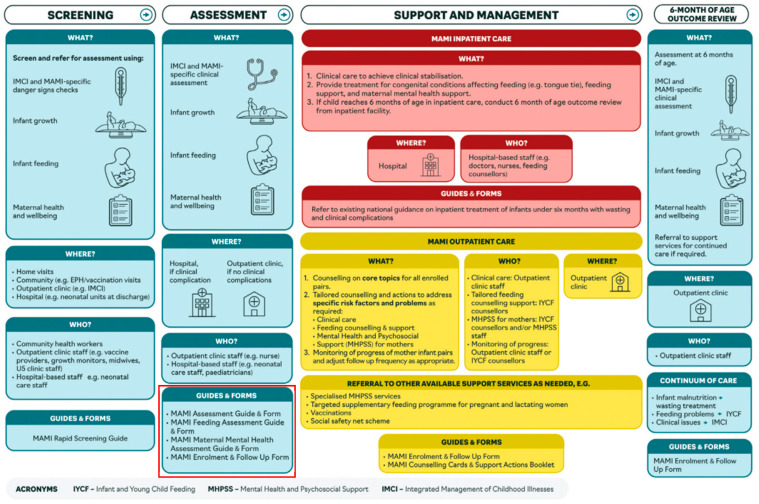
Summary of the MAMI Care Pathway [22]. The red box highlights one example of how MMH (under the overall label of “maternal health and well-being”) can be integrated into infant feeding assessments and care.

## 2. Materials and Methods

### 2.1. Study Design

This is a cross-sectional qualitative study using semi-structured, in-depth interviews with international key informants working on nutrition programme implementation, programme management, or mental health research, some of which has been used to inform international policy. We chose this key informant study design because it is a well-established methodology to rapidly but thoroughly explore an issue and generate hypotheses and understandings which can be used to inform policy, practice and future research [24].

### 2.2. Participant Selection and Recruitment 

Eligible participants were defined as having a minimum of one year’s experience in the field of nutrition and/or mental health programming or research for policy making, working in LMICs. This criterion was to ensure participants had sufficient knowledge of the key clinical practice area and relevant experiences to discuss. There were no criteria regarding age, gender or previous work experience, as the intention of this paper was to collect data from a range of participants, not a specific subgroup. 

Participants were initially selected via convenience sampling from a professional network of nutrition research and programme experts. Further participants were identified via a literature search to find professionals who had collaborated on the mental health element of the MAMI Care Pathway, collaborated on the WHO guidelines or contributed to published research regarding MMH assessments and infant nutrition and were health practitioners. Snowball sampling [25] was adopted following the initial recruitment period after several nonresponses to recruitment emails (Figure 2). This involved participants recommending others in their own professional networks. 

#### Sample Size

Data saturation was used to obtain an appropriate sample size for the study [26]. For this study, saturation was deemed as a process, and the intentions of the authors were to reach a high level of saturation in which participant responses may differ due to different working contexts but key ideas and concepts are repeated. Building on past experiences from similar work, a target of 10–15 participants was chosen.

### 2.3. Data Collection

#### 2.3.1. Interview Guide 

The data collected from a 2022 online quantitative MMH/infant malnutrition survey were used to inform the first draft of our interview questions [27]. 

Two pilot interviews were conducted allowing for the first draft of the interview guide to be field tested. The interview guide is available in the Appendix A.

#### 2.3.2. Conducting Interviews

Interviews were arranged with prospective participants by email. The interview questions were not provided to participants prior to the interviews. This was a desk-based project, as participants were recruited from all over the world. All interviews were conducted via Zoom, and only the interviewee and N.M. were present in each interview. 

Interview audio and video were recorded to assist with transcription. Where possible, web cameras remained on during the interviews. No repeat interviews were required, as all interviews were successfully recorded. 

Prior to the start of the interviews, participants read a study information sheet and provided informed consent to take part. Because participants were fellow health professionals and we were asking about professional opinions rather than personal issues, the study was low risk from an ethical perspective.

#### 2.3.3. Transcribing 

The aim was to complete the transcription of a single interview within 48 working hours after the event to allow for any queries to be highlighted and addressed as soon as possible. However, the time taken to complete the transcripts varied depending on the length of the interview and N.M.’s availability between other scheduled interviews. 

Verbatim transcripts are not required for a thematic analysis, thus nonverbal communications and interruptions to the interviews were not included. Some amendments were made to correct obvious grammatical mistakes to allow for accurate and clear data analysis. We did not return the transcripts to participants once dictated for confirmation of the text due to the prompt manner in which transcripts were completed following the interviews. 

All data were stored using the encryption software VeraCrypt (version 1.25.9) in encrypted, password-protected files. Only the final, anonymised transcripts were kept. 

### 2.4. Data Analysis

A thematic analysis was selected as the data analysis method to detect and examine themes and subthemes within the data [28]. This was chosen because we wanted participants from varied backgrounds and wanted to compare views and experiences in each of the key topic areas.

Once the transcripts were completed, they were re-read to confirm accuracy. Extracts from each interview were grouped based on participants’ responses to certain questions. The interview guide was developed to generate data regarding four predetermined themes, which were identified from the previous MAMI survey [27] and this study’s research objectives. The themes were perceptions and understanding, challenges, benefits and risks, and improvements and innovation. 

With the data organised into the four themes, the transcripts were imported into the NVivo12 programme. N.M. coded the transcripts, identifying the subthemes and using these to code and organise the data; as this was an inductive approach, the subthemes were not predetermined. Within the four key themes, more than 30 subthemes emerged (Table 1). The references within each of these codes were analysed and compared. 

### 2.5. Ethical Approval 

This study was reviewed and approved by the London School of Hygiene and Tropical Medicine Ethics Committee (Reference: 29028) prior to the commencement of any research. 

## 3. Results

### 3.1. Sample Characteristics 

Table 2 depicts the characteristics of the 10 study participants (P1–P10). Data were collected during pre-interview questioning. Participants 1, 8, 9 and 10 have all conducted research to inform regional-, local- or international-level policy or practice to different degrees. Participant 4 was a manager of their programme at a sub-regional level, and they were listed as an implementer, as Participants 2, 3, 5, 6 and 7 worked at a larger, regional level. Most participants (70%, 7 of 10) worked in Sub-Saharan Africa; 30% (3) worked in Asia.

### 3.2. Themes and Subthemes 

Figure 3 summarises the relationships between the themes and subthemes. 

The most frequently discussed subthemes are presented, as these topics have clear importance within the larger themes. There were additional subthemes that were closely linked to these key topics, which are alluded to where they add important detail and context to the findings.

#### 3.2.1. Perceptions and Understanding 

##### Mothers’ Perceptions 

Not all participants worked directly with mothers in their settings; therefore, not every participant was confident they could respond to this question. There was some variation in the reporting of mothers’ perceptions of MMH and assessments. Some participants reported that mothers were concerned about the consequences of screening positive on mental health assessments, for fear of being labelled “mad” or having their children taken away from them. P6 discussed their experience of mothers’ perceptions:


*“I see the mothers are less convinced, less informed about the relevance of those screenings… I mean they have an issue of trust with health professionals, because they fear judgment… They fear stigmas, discriminations”.*


The importance of mothers understanding MMH assessments was also highlighted. P5 explained that, additionally, mothers can feel frustrated at being asked about their mental health when attending an appointment for their child and have difficulty conceptualising mental health and explaining it to their family. Hence, most mothers respond “No” to questions about their mental health symptoms. 

Some participants highlighted that mothers in their settings were advocating and asking for more mental health support. For example, P4 and P7 both work with displaced populations and P4 reported that women were pleased to receive mental health interventions. P7 explained that the mothers, in their context, were asking community healthcare workers for more mental health support and services, as they identified that they were struggling with their circumstances. P1 advised that mothers seemed open to discussing mental health in community health clinics; however, they felt unable to do so because of concerns over how they will be perceived by others. 

##### Staff Perceptions 

P6, P7 and P8 explained that staff generally felt positive about assessing and screening MMH. However, P6 stated that, although staff were happy to learn and adopt the new skills, the implementation of these skills was inadequate. P6 explained that, in practice, healthcare professionals were not exploring the presence of MMH symptoms, merely asking the screening tool questions without engaging the mothers in actual discussions regarding their mental health. P1 found a similar situation when staff were taught to detect maternal depression; despite teaching midwives content from the mhGAP, the rates of detected depression did not change, as P1 disclosed: 


*“…midwives, repeatedly found that it had no impact on detection of depression at all… there was a reluctance for the providers to ask... they said: oh the women don’t want us to ask… it also included a sense that mental health problems were due to external circumstances… So there was, sort of, limited sense of: why would we ask about these things when they’re social and we’re not going to do anything?”*


Additionally, P10 explained that staff may make assumptions about some mothers, thus choose to not ask them certain screening questions.

##### Community Perceptions and Stigma 

Almost every participant commented on negative and potentially harmful community perceptions and stigma, including judgements held by mothers themselves against MMH and screening assessments, which can create additional pressure on the mother. Several participants offered potential explanations for these stigmas and beliefs. P2 advised that in traditional communities in their setting, religion plays a significant role in influencing community beliefs and understanding of mental health. Thus, families seek advice on exorcism before considering healthcare. P9 also reported on the influence of religion and tradition, explaining that mental health conditions are believed to be caused by black magic or possession. Thus, the community would expect a mother to initially seek help from a spiritual healer, which, as P9 explained, then leads to a delay in interventions from medical professionals. Conversely, P5 advised that clinical psychologists are viewed as unimportant by the communities they work with, so discussing mental health is unacceptable to these societies. Furthermore, P5 explained that, in some areas, there is a lack of importance placed on mental health, as difficulties are a part of life.

P9 discussed the improvements family members observe after MMH screening and treatment, which greatly improves the acceptability of these assessments:


*“When they see the improvement in the mother, they feel happy… the husbands, when they see their wives getting better, looking after the personal hygiene, less arguments with the husband. So they really appreciate the program… likewise the mothers-in-law and the other family members”.*


#### 3.2.2. Challenges

##### Governance 

The most commonly reported challenge to the effective implementation of MMH screening was governance. This was defined as a lack of systems in health settings, poor health service structure, a lack of clinical governance for practice in maternal healthcare, a lack of policy, and a lack of clinical care guidelines. Most of the participants (P1–P8) reported facing challenges with governance. P5 had many example experiences to support this sentiment. P5 stated that, whilst mental health is becoming more of a priority in their setting, the pace of development is slow, especially without government backing and advocacy, and the pace of implementing MMH services, in practice, is slow. P1 and P8 advised that without the right health systems in place, there is nothing to keep staff accountable or prompt them to screen and assess MMH. As P8 points out: 


*“If it’s not part of their key performance area in their job description, if they’re not being monitored against the screening… if there isn’t enough training for that to happen and if there aren’t any resources… screeners feel well, why should I screen? I think there’s an ethical responsibility to have resources in place, care pathways in place and one that you trust”.*


##### Staff Training and Capacity 

Every participant commented on challenges with staff training, skills and capacity to implement screening assessments. A key challenge that was highlighted was high staff turnover, which was particularly testing in P2’s and P3’s work settings. P4–P8 all mentioned how the time staff have available to effectively and safely implement screening assessments is entirely inappropriate and insufficient. P5 explained that integrating mental health screening into infant feeding assessments is very difficult for their staff, who have up to and sometimes over 100 patients per day. In P7’s setting, mothers are in displacement camps, which have strict regulations on access and how long health professionals can be present in the camps: 


*“So if staff have like an average of three to four hours a day to be in contact with the mothers, and then also the different activities, it doesn’t really give ample time for them to, you know, to do the assessment, to do the counselling”.*


In addition to struggling with time, key problems participants experienced included staff training and skills to implement MMH screening assessments. The challenges varied from not having sufficient training uptake to inadequate numbers of trained staff and the scarce provision of training at various points during career development. P4 explained that the health-implementing NGO partner in their setting offers mental health training to the nutrition service staff, which they do accept, but there is no specific training offered in-service. However, P6 described that external training is not enough to instil confidence in the nutrition service staff. P1 felt that one solution to this issue could be more training at the grassroots level. 

##### Finance 

Almost half of the participants reported on finance and funding. P3 and P7 commented on how mental health assessments and interventions are not prioritised in budgets; however, money was given to mental health programming, providing it was integrated into other services. On the one hand, this may make logistical sense, especially in resource-constrained settings, wherein it may be more convenient to integrate MMH assessments. Alternatively, it could be a reflection of how mental health is not considered an equal health concern in need of attention. The latter point may be further supported by P4:


*“Funding by itself is one of the challenges, if there is no available fund to recruit enough staff for the project… it’s also one of the challenges that we are facing”.*


However, P5 felt that funding is not the biggest challenge they are facing in their context, and P2 also felt that given their setting’s circumstances, although budgets are a huge consideration, their service is offering as much as they can to their beneficiaries. 

#### 3.2.3. Benefits and Risks 

##### Health, Wellbeing and Suicidal Intent 

Many participants commented on the therapeutic benefits mothers may experience from mental health assessments. It was mentioned several times that particular phrasing of mental health questions integrated into infant feeding assessments could yield benefits. So health workers were not directly asking about common mental health disorders but asking general questions that could uncover symptoms of an MMH condition or external factors that could be causing stress, which, in turn, could have implications for mothers’ wellbeing and, subsequently, improve the care of infants. P6 explained that in their setting, MMH screening is conducted by a trained professional who understands MMH conditions, which allows the mothers to discuss their situations and feel understood by the care staff. This, in turn, helps to maximise the benefits of improved maternal wellbeing. P4 and P7 commented on the benefits to the wellbeing of the whole family/household of discussing MMH with mothers, and P7 highlighted the importance of trying to capture this as an outcome in programmes. 

However, some felt that with inadequate training and supervision, screening all mothers could be harmful, as it could lead to a misdiagnosis and the mismanagement of mental health problems. P5 explained that whilst you can implement an integrated screening tool into an infant nutrition programme and have all the resources and services available, if this MMH screening assessment is not conducted adequately, an at-risk mother–infant pair will not receive any benefit from this process. Furthermore, P5 advised that as MMH is not widely discussed and addressed in the communities they work with, care staff risk losing mothers and infants to follow-up, as mothers feel discussions over their mental health, especially in a hospital setting, are not related to the health and nutrition of their child. So mothers see no benefit in travelling to these appointments for treatments; therefore, they stop attending, which can have obvious ramifications on infants’ health and nutrition. 

Multiple participants also felt there was an ethical responsibility from the service provider to have a mental health support service if they intended to screen mothers. Otherwise, some felt there was no benefit to screening if there is no subsequent support. This was a common discussion point when participants were asked about suicidal intent too, as P10 explained: 


*“If you don’t have any referral, available system of any kind, it might be unethical to ask about suicidality. If you’re planning to take the fact that the answer is yes, and then do nothing with it, because you may do more harm than good, because the woman has then told someone and had no reaction, which might lead her to feel no one cares”.*


Some felt it was acceptable to ask mothers about suicidal ideation during a mental health screening providing the wording of the question was sensitive, appropriate for the context and asked in a culturally acceptable way by an appropriate health professional. A number of participants also explained that even in a low-resource setting, there should be minimal risk from asking mothers about suicidal thoughts and plans, as there are some emergency services that may be able to support mothers with suicidal ideation. Additionally, the risk of harm may be lower in some settings in which it is culturally unacceptable to end one’s life. 

#### 3.2.4. Improvements and Innovation 

##### Staff and Service Development

Service pathways and staff development were identified by participants as some of the most significant points for improvement and innovation. Service changes such as the integration of MMH assessments into existing services was highlighted for cultural acceptability, as well as feasibility in a low-resource setting. P2 explained that they have been building onto services to introduce mental health treatments, so there was better continuity, wherein the mothers are screened at the health clinic and then guided into mothers’ groups.

Clearer referral pathways were identified as a need in some settings, which relates to the ethical responsibilities of service providers. Staff require better communication regarding how to refer to specialists, as well as what services are available in the locality, particularly for displaced populations, to ensure resources are still available for the host communities. 

Staff training was frequently mentioned as an area for improvement. Suggestions for improving staff skills included better grassroots training, more supervision, more training opportunities and more accountability for staff. Some participants shared the training they offer the staff with who they work. Several participants reported that they were already trying to work on improving staff skills without increasing training requirements by developing more efficient screening tools. P8 provided details of their work: 


*“When we developed our own tool, we wanted it a) to be brief, for the scoring system to be practical for a busy health worker... We wanted the construct to be binary—yes, no—and we wanted few items as possible… a first layer of screening… You’ve tested that, you’ve done cognitive testing, as well as psychometric testing on that, and then you move on to a more in-depth assessment of some form”.*


##### Community and Tradition

The use of community bonds, tradition and indigenous knowledge was another common topic raised by participants. Many felt that MMH screening could be strengthened at the community level, whereby community health workers are used to monitor mothers and infants and then inform health professionals if they are concerned about any at-risk families. This was suggested by P1 and P7 as something that could help in their settings, and P4 explained that their team is currently implementing this concept. P4 advised that they had recruited incentive workers, who are local women from the displaced communities, to implement basic screening of malnutrition and MMH. 

P2 and P3 stressed the importance of community in many societies in LMICs, and explained that these networks and systems should be facilitated and supported. This would be useful for identifying at-risk cases, as suggested and explained above, and may provide space for mothers to discuss mental health concerns in a culturally acceptable way. This has the potential to be more clinically effective and improve maternal and infant health outcomes. P2 describes: 


*“We try and strengthen natural support system. So yeah, coffee ceremonies, local communities, re-establishing local communities, even community centres, if they exist, even if it’s within the context of the church. But those systems actually have a purpose and are really helpful to people… promoting that within this community and creating awareness, we can do in different ways, in groups or larger activities as well, to say: it’s OK not to be OK”.*


Several participants discussed the need to improve community mobilisation in their settings for several reasons. P8 had implemented some steps to help improve mothers’ perceptions of mental health, such as including a standardised psychoeducational elements to their care pathway to help dispel some community beliefs. P6 expressed the importance of working with communities to tackle stigma and normalise discussing MMH; otherwise, they felt, any other changes to improve MMH screening would likely be unsuccessful. P3 explained about using community leaders and influential members to make assessments more feasible and acceptable. 

##### Policy and Investment

Funding and governance were reported as challenges to implementing MMH assessments; hence, investment and policy were commonly reported as areas for improvement and innovation. Investment extended past securing more money; as P4 explained, it is also important to consider investment in staff, training and resources to better implement MMH screening assessments and support: 


*“…but the challenge is not having enough budget or well-trained personnel in one area. In the future, if we fulfil these things, I think it is very supportive and would improve the quality of the service to the community”.*


Additionally, in order for the suggested improvements and innovations to happen, P3 felt that national-level policy needed to change, starting with improved advocacy.

## 4. Discussion

This study found that no matter the challenge, risk or area for improvement, governance, policy and advocacy are consistently identified as limiting the feasibility and acceptability of assessing MMH, when managing nutritionally at-risk infants. Stigma and perceptions of MMH are key barriers to the acceptability and feasibility of MMH assessments, but community involvement in programme planning and implementation may help to overcome some of these difficulties, particularly by providing mothers and communities with locally/culturally acceptable services.

### 4.1. Perceptions and Understanding 

Whilst there was no overall consensus on maternal perceptions of MMH across the different contexts, it is evident that circumstance, understanding of MMH, wider health service set-up and care delivery systems all play important roles in formulating mothers’ perceptions of MMH assessments, either positively or negatively. Half of the participants who discussed staff perceptions state that staff are accepting and willing to screen MMH at nutrition assessments. In spite of this, most responses included some negative perceptions of mental health. There were also examples of how this subsequently influenced staff clinical practice, such as poor implementation of MMH assessments. Both maternal and staff perceptions could be related to the overall communities’ understanding and beliefs regarding MMH, thus establishing the basis on which their perceptions were made. A similar conclusion was found in a cross-sectional study in Sweden [29]. Despite Sweden being a high-income country (HIC), other data from a global review of patient experiences of being treated for mental health conditions showed many similarities in attitudes and discriminations faced in the different settings [30]. Thus, whilst local context varies and MMH-specific beliefs vary in different in HICs and LICs, the discrimination and stigmas are still present in all societies. The findings of this paper support this conclusion. The need to reduce stigma in LMICs is also well documented [31]. Previously used mental health antistigma strategies can be adapted and implemented [32]. 

From the interview responses, the general feeling was that community views and stigma against MMH was common in almost all settings. When considering how effectively MMH assessments can be implemented into infant nutrition programmes in LMICs, perceptions and understanding are important factors. They can influence the cultural acceptability of screening assessments and the feasibility, for instance, engagement from care providers, mothers and families, and, thus, the ability to integrate into the health system. This conclusion has also been drawn by a key informant, who contributed to a recent landscaping analysis regarding the state of MMH in LMICs [33]. 

### 4.2. Challenges 

The responses showed that there are clear, cascading links between policy and government backing, clinical guidelines, health service design (including training and supervision) and implementation of screening assessments by health professionals. Without strong policy support from governments, there is limited scope to feasibly and successfully implement systematic screening programmes. This is not a novel finding; a recent review of global literature on mental health surmised that a key step for the long-term improvement of mental health in LMICs was identifying potential changes to government policy [31]. These data were not specific to MMH, and it is important to distinguish among the more specific MMH difficulties patients may face. In most of the LMIC settings discussed in this paper, there is limited advocacy to include MMH education in staff training; limited drive to hire more staff to better distribute caseloads and encourage better quality MMH screenings; and a lack of clinical guidelines, which would support integrating and implementing MMH assessments. These factors directly relate to weak governance, which is evident across a range of contexts. Given this context, it is even more encouraging that new WHO malnutrition guidelines do highlight MMH numerous times in the text [20]. It remains however to be seen how this global-level recommendation will be implemented at the country and local levels. Previous experience with another often neglected topic strongly linked with nutrition—disability—show that not all international recommendation successfully filter down to local levels [34]. Such omissions during the rollout and implementation process need to be avoided if MMH work is to succeed. 

Whilst funding may be perceived as a key barrier in wider public health nutrition programming in LMICs, participants in this study had varied thoughts on this. Recent literature highlights funding and investment as key challenges in improving MMH care [32]. There could be a bias within this observational commentary, but the article does align with opinions from participants in this study, suggesting validity in the report. The challenge is, as P2 reflected, that if more money was available, more could be done to make MMH assessments feasible in LMICs. However, adequate funding is simply not available in many resource-constrained settings. Even authors of a model system for optimal MMH screening and treatment in LMICs agree that to implement and sustain such work in practice, there needs to be an increase in spending, and this is difficult in some settings [33]. Whilst finance can be a limiting factor in true clinical efficacy, service design and delivery (i.e., governance) can be a bigger hindrance to more widespread MMH screening. This is also true for wider public health and nutrition issues. Political priorities and clinical/public health needs do not always coincide. 

### 4.3. Benefits and Risks 

There was no unanimity on the health and wellbeing benefits or risks to mothers from MMH screening. Many points were raised about the benefits for mothers, infants and their families, but, equally, many concerns were highlighted about the potential for harm. Screening might, for instance, raise expectations and leave mothers feeling even more unsupported if adequate follow-up services are unavailable. This should be seen in the context of some available evidence. A recent meta-analysis and systematic review found moderate evidence to show that screening, in fact, does have moderate evidence of a positive impact on depressive disorders and high-quality evidence of improvement in anxiety symptoms [35]. However, the studies in that review were not from LMIC/malnutrition programmes but mainly from HICs where follow-up was available. Concerns regarding the possible harms of MMH screening are evidently justified and data are urgently needed on settings that better reflect typical malnutrition programmes in LMIC settings. It is plausible that even the simple act of a healthcare worker asking about and caring about MMH, as well as treating mothers with the kindness and sensitivity that all deserve, is therapeutic. However, it also plausible that, in some cases, harm may result if adequate follow-up is unavailable. The acceptable risk/benefit balance is likely to differ in different settings. Because there are multiple later impacts of MMH on infant feeding, nutrition and health, additional public health nutrition outcomes should be considered, accounted for and measured when integrating MMH screening assessments into infant nutrition care pathways. 

The views on suicidal questioning were equally varied, with many in favour of screening and some against this being routine. The key issues leading to perceptions of benefit or harm were related to phrasing, cultural acceptance and gradation, wherein questions are asked only if indicated by the mothers’ other responses. There was a focus on the ethical implications of asking mothers screening questions if there was suboptimal support to offer them should they screen positive. Wider evidence, again, suggests that screening and asking about suicidal intent is itself important and does not increase risk [36,37,38]. As before though, these data are mainly from HICs and evidence from LMIC/malnutrition programme settings is urgently needed. Both sets of views pose important considerations for policy and programme designs. On the one hand, it is important to ask about MMH since documented need is a vital first step to justify future treatment services; on the other hand, services are needed to safely manage cases if identified. Once again, situation-specific governance and advocacy could determine the right balance between the potential for benefit and potential for harm in different settings. Solid governance is vital for strong MMH care systems in a country or region, and our study is not unique in highlighting that policy makers and programme specialists need to be more aware of mental health issues and improved governance needs to be set up to facilitate effective and safe services [39]. There are numerous, well-documented ways forward, including task-shifting, which could help strengthen services in the short and medium terms [40]. 

### 4.4. Innovation and Improvements

Participants identified potential areas for improvement in clinical practice, particularly staff development and service pathways, with some participants demonstrating ways in which they have responded to these needs. Some participants had designed more efficient MMH screening tools for staff in their settings, so there may be scope for larger scale implementation in the future. These include the screening approach promoted by the MAMI Global Network in the MAMI Clinical Care Pathway for small, nutritionally at-risk infants u6m [41,42]. Although MMH specialist staff may be a valuable investment, many participants felt that MMH specialists are not necessary at all stages of care pathways for effective implementation of MMH assessments. This is consistent with other evidence that task-shifting to more generalist staff can be effective [40]. To further boost the chances of success, there is also a need for ongoing staff training and for better grassroots-level training for health professional qualifications. Some data support better mental health training for student health professionals as a good way to reduce the stigma and prejudice they may hold as qualified professionals. For example, clinical placements in mental health have been shown to have many benefits, including “improving students’ skills, knowledge, attitudes towards people with mental health issues and confidence, as well as reducing their fears and anxieties about working in mental health” [43]. These data were not specific to LMIC settings and, thus, should be extrapolated with caution, but the key principle is that more and better MMH training for future and current healthcare staff has great potential internationally. This could result in improved staff understanding and skills related to MMH, with benefits not just for mothers but for infant feeding, health and wellbeing. Integration with multiple downstream benefits has been successfully implemented in other health areas. For example, there has been strong governance and investment into the integration of the Prevention of Mother to Child Transmission of HIV (PMTCT) programme in Maternal and Child Health Services [44]. Work included training, capacity building, screening, counselling and treatment; all this has yielded significant positive results in the reduction of HIV cases. There have also been good experiences of integrating MMH into routine maternal care service [45]. Such impactful learning experience could be utilised by the nutrition community to help influence governments and donors to invest in the integration of MMH into infant nutrition programmes. 

Participants discussed using mental health specialists for support at a higher systems level, supervising lay staff in complicated screening assessments. This is important for programme design and funding, as without budgeting for specialist input, MMH screening may not always be safely implemented. Some existing studies examine the positive effects of task-sharing and supervision of lay staff in delivering MMH interventions [40]. These data focus on wide-ranging interventions rather than MMH screening/assessment alone, but they do clearly show that there is scope to better utilise staff skill sets and invest in adequate training and supervision of lay staff to implement MMH screening. 

Many participants commented on the utilisation of community networks and indigenous knowledge as a practical and effective method of improving the feasibility and acceptability of MMH assessments and interventions in LMICs. This has been a consistent finding in different pools of evidence; for example, the utilisation of Nutrition Impact and Positive Practice (NIPP) approach for the improvement of child undernutrition through positive deviation [46]. Specifically relating to mental health services, a systematic review highlighted the importance of framing and prioritising mental health services in light of “local population and cultural needs” [47]. Frameworks such as ADAPT exist and can be used to help guide such local adaptations [48]. This is another example of how there is scope for MMH assessments to be effectively implemented in future malnutrition programmes. 

The responses also indicated needs in multiple areas of programme management. Whilst funding without advocacy may still yield some benefits, most of the changes and improvements suggested will struggle to be successful without investment, political backing and a more positive, supportive shift of mental health in political landscapes in LMICs. 

### 4.5. Strengths and Limitations 

Qualitative studies may be regarded by some as low-quality evidence. This is only true if the aim of a study is to determine causal relationships between a factor and an outcome [43]. In contrast, our paper aimed to generate an understanding and hypotheses on the acceptability and feasibility of MMH assessments when managing nutritionally at-risk infants in LMICs. Therefore, our data offer a valuable starting point and important insight into this topic, which we hope others will pick up on in the future. We also deliberately frame our work as an “acceptability/feasibility” study—this is designed to inform future GRADE guideline reviews that consider qualitative, as well as quantitative/effectiveness, data [49]. Another strength is that we add to a field that otherwise has very limited, high-quality published data specific to MMH and infant nutrition programmes. Through speaking with key informants, we obtained data from international experts in the fields of child nutrition, mental health research and nutrition programme management. Thus, the quality of discussions and accounts from all participants is a significant strength of this study, made feasible by the semi-structured interview methodology. 

We also acknowledge limitations. First, there is some homogeneity in the study sample, with recruitment revolving around professional networks and snowball sampling and, thus, not fully representing the wider public health nutrition community. At this stage, such a selection bias problem is difficult to avoid, but we hope that others will apply our frameworks and questions to wider samples in more geographical regions. Even though some of our participants worked in the same countries, or even for the same organisations, there were still differences in the responses. This suggests value and variety of perspective despite how participants were recruited. Additionally, whilst the sample is not necessarily representative, and some could argue a consequent limited applicability of our findings, on some issues there is a strong element of conceptual generalisability, wherein the larger themes and commonly discussed subthemes are applicable to a wider public health nutrition setting despite local variations. This also shows that a high level of data saturation was reached in this study. 

The data collected add depth and detail to the previous quantitative MAMI survey data, which identified variables affecting MMH assessments in LMICs when managing small, nutritionally at-risk infants [28]. This study has allowed for a greater analysis of these themes, so the wider public health community may better understand the barriers, benefits, risks and future improvements needed to ensure MMH assessments are more acceptable and feasible within the context in which they work. An alternative methodology would not have allowed this study to reach the same level of detail to compare and create a hypothesis. An alternative recruitment method could have been adopted to avoid any possible sampling bias; however, one could argue that given that people working in the same contexts provided different responses to the questions, it would be unlikely that an alternative methodology would have yielded significantly different results. 

### 4.6. Recommendations

To build on our work and to continue to improve MMH/malnutrition programme integration, we highlight the following future research priorities: To align with the release of the new WHO malnutrition guidance, longitudinal studies should examine the process and uptake of MMH screening in infant nutrition programmes. Such studies could include patient and staff questionnaires on acceptability of different approaches taken. Surveys could quantify and more definitively answer some of the questions we raise (e.g., “Does MMH screening improve mental health and nutrition outcomes in the absence of a specialist treatment service?”; “Is asking about suicidal ideation acceptable and is there any evidence of increased risk associated with the question?”; “Which assessment tool works best in which setting(s)?”Analysing the return on investment of MMH screening and care pathways.By collecting data on the short- and long-term cost-effectiveness of acceptable screening and MMH care, future programmes are more likely to attract donors and could draw more government support. Economic data would resolve many of the uncertainties raised in the current study and would help make a case for future investments (if, of course, the results indicate cost-effectiveness). Data specific to LMICs are needed to examine the relationship between mental health training for health and medical students, as well as subsequent attitudes in clinical practice. This would be vital for influencing health workers’ attitudes towards mental health in LMICs, as well as for improving communication skills to better deliver the MMH assessments.

Research is also needed to highlight the burden of MMH in individual settings to better lobby and advocate for more support and funding from governments and policy. However, it is important that this research is conducted ethically, without putting mothers at risk. Therefore, we advise the following for research and clinical practice in LMICs: Community mobilisation and cultural adaptation of any MMH screening is fundamental to ensuring the acceptability of the screening. Communities should be involved in the planning and delivery of MMH screening and treatments and, where possible, these should be integrated into existing services but primarily into existing social norms and support networks. This ensures long-term sustainability and is likely to make the assessments more feasible and acceptable to the target populations.Ongoing staff training, sensitisation and supervision are needed to better support both the staff and the mothers they are treating. With community and staff perceptions being largely negative, it is important to develop an understanding of MMH and ensure mothers and staff are engaged in screening assessments and that no harm comes to mothers or their infants.Service pathways need to be more robust, clearer for staff to understand and, ideally, not add too much work onto the roles of lay staff.Donors could play a role in normalising MMH in infant nutrition programmes by making MMH integration a criterion to access funding. This would enforce accountability of service providers to integrate MMH into service designs. Data would need to be collected on this element of care, which could add to the evidence base, and this would encourage nutrition programmes to follow the WHO’s guidance. However, there needs to be strict guidance on including MMH in programmes, wherein support services are also provided.

Screening MMH can benefit the mother and, subsequently, their infant, but the data from this study clearly emphasise that providers have a duty of care to offer support to the mothers who screen positive for MMH conditions. This must be a key element of future programme implementation. This is not just a matter of addressing a major—yet largely unmet—current need, but, as some authors rightly frame things, an issue which is integral to essential child rights [50].

## 5. Conclusions 

Our study suggests that no matter the challenge, risk or area for improvement, governance, policy and advocacy are major factors hindering (or enabling) the feasibility and acceptability of MMH assessments when managing small, nutritionally at-risk infants. These should be key focus areas as governments, NGOs and others seek to roll-out new WHO malnutrition guidelines that now link MMH and malnutrition. Stigma and negative perceptions of MMH are key barriers to acceptability and feasibility of MMH assessments, so community involvement in programme planning and implementation is vital to help to overcome this. Finally, more evidence from LMIC/malnutrition programme settings is urgently needed to justify and promote widespread MMH screening. In the immediate future, there are many steps that current service providers in low-resource settings can undertake to improve the care they offer mothers and infants. This is vital from a rights perspective, as well as to meet an important and currently unmet clinical and public health need.

## Figures and Tables

**Figure 2 children-11-00209-f002:**
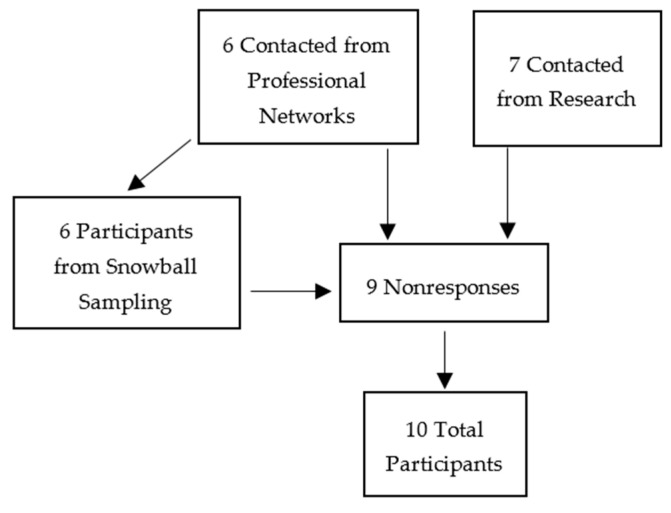
Flowchart of participant recruitment.

**Figure 3 children-11-00209-f003:**
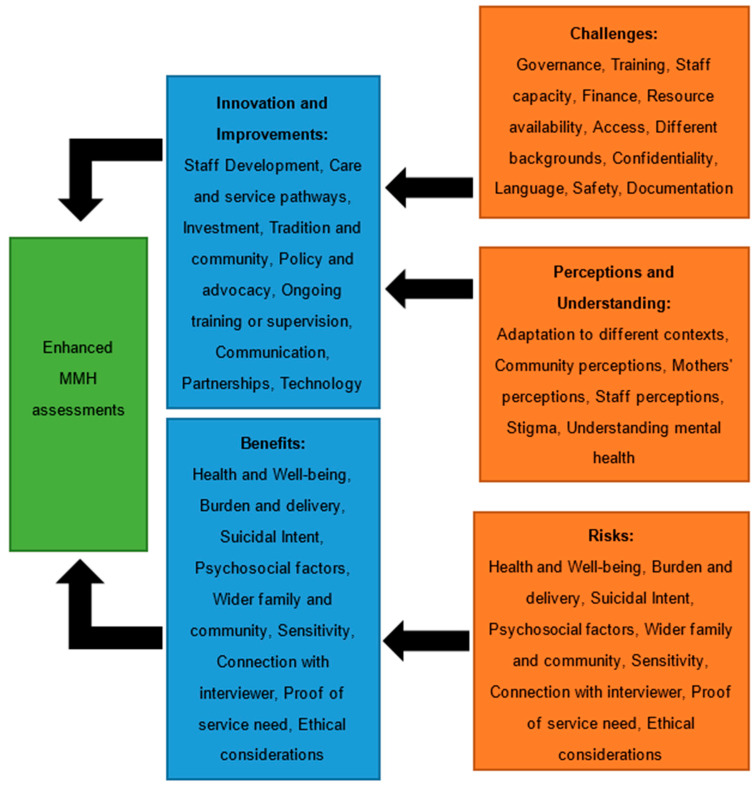
A framework illustrating the relationships between the key themes and subthemes identified in the data, feeding into enhanced MMH screening.

**Table 1 children-11-00209-t001:** Codebook depicting the number of codes per overall theme.

Theme	Titles of Codes Identified	Number of Codes
Benefits and Risks	Health and wellbeing	15
	Suicidal intent	11
	Psychosocial factors	8
	Wider family and community	8
	Sensitivity	7
	Connection with interviewer	6
	Proof of service need	3
	Ethical considerations	2
Challenges	Governance	27
	Training	16
	Burden and delivery	14
	Staff capacity	13
	Finance	12
	Resource availability	11
	Access	10
	Different backgrounds	6
	Confidentiality	5
	Language	5
	Safety	3
	Documentation	1
Improvements and Innovations	Staff development	25
	Care and service pathways	23
	Tradition and community	18
	Investment	17
	Policy and advocacy	14
	Ongoing training or supervision	9
	Communication	6
	Partnerships	4
	Technology	3
Perceptions and Understanding	Adaptation to different contexts	14
	Community perceptions	13
	Mothers’ perceptions	11
	Staff perceptions	10
	Stigma	7
	Understanding mental health	6

**Table 2 children-11-00209-t002:** Sample Characteristics.

Participant	Current Role	Current Region of Work	Additional MMH Training	Input with MAMI Care Pathway
P1	Researcher informing policy	Sub-Saharan Africa	No	Yes
P2	Nutrition programme manager	Sub-Saharan Africa	No	Yes
P3	Nutrition programme manager	Sub-Saharan Africa	Yes	No
P4	Nutrition programme implementer	Sub-Saharan Africa	No	No
P5	Nutrition programme manager	South Asia	No	Yes
P6	Nutrition programme manager	Sub-Saharan Africa	Yes	Yes
P7	Nutrition programme manager	Southeast Asia	Yes	No
P8	Researcher informing policy	Sub-Saharan Africa	No	No
P9	Researcher informing policy	South Asia	No	No
P10	Researcher informing policy	Sub-Saharan Africa	No	No

## Data Availability

The data presented in this study are available on request from the corresponding author. The data are not publicly available due to the need to maintain individual respondent confidentiality due to sample size.

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
