# Peer review of "Acceptability and Feasibility of Maternal Mental Health Assessment When Managing Small, Nutritionally At-Risk Infants Aged < 6 Months: A Key Informant Interview Study†"

_children, 2024, doi:10.3390/children11020209_

Round 1
Reviewer 1 Report
Comments and Suggestions for Authors
This manuscript is important and provides a useful roadmap for future efforts. The figures were particularly helpful and concise. Because this is a worldwide effort, some information on where the 10 total participants reside and/or a summary of their training ane experience would be helpful.
Comments on the Quality of English Language
Some abbreviations, like row 51 "u5", are not explained. Assume that means under 5 years of age but would be helpful to detail.
Author Response
Reviewer 1
Comments and Suggestions for Authors
This manuscript is important and provides a useful roadmap for future efforts. The figures were particularly helpful and concise. Because this is a worldwide effort, some information on where the 10 total participants reside and/or a summary of their training ane experience would be helpful.
Response
- We thank the reviewer for this supportive and helpful review. We likewise hope that jounral readers and researchers will find our paper helpful for future efforts.
We have noted “region of work” and “maternal mental health training” (including experience with the MAMI care pathway) in table 2 describing participant characteristics. This is the core data needed to understand the profile of respondents. Due to the small sample size and associated need to preserve anonymity of our respondents we did not collect more granular information about residence and training. We are fully aware that larger numbers of respondents from a larger number of different locations and with different personal/professional backgrounds may give additional or different insights and we thus, in both abstract and main text.:
- Are cautious in our conclusions and do not overinterpret what these 10 respondents alone say / do not take these to be necessarily representative of a much wider group
- Call for further research to address this issue of wider generalizability.
Comments on the Quality of English Language
Some abbreviations, like row 51 "u5", are not explained. Assume that means under 5 years of age but would be helpful to detail.
Response
Thank you for flagging this. We have gone though and explained abbreviations at first use, including this one.
Reviewer 2 Report
Comments and Suggestions for Authors
Thank you for this interesting article.
The article addresses an important health concern in maternal health. Through key informants as the participants, this article set-out to determine the feasibility, acceptability, and perceived benefits and risks of asking about Maternal mental health (MMH) as part of the routine assessment for infant malnutrition, especially in settings where specialist MMH services are unavailable, limited, or difficult to access.
Thank you authors, for taking the time to put this piece together. However, there are a few issues that you need to look into. Below are my specific comments.
Abstract- This section is fairly written, with the major components of a good abstract included. This section can pass in its present form.
Introduction:
-Kindly insert a citation in line 34.
-Remove data collection methods mentioned in the study design, i.e., semi-structured questionnaires, in-depth interviews etc. take them to data collection methods. Justifying why you chose a cross-sectional design and not others will also be essential.
-Which sampling strategy led to 10 participants in this study? Additionally, describe how you conducted snowball sampling and its relevance.
-Information in lines 181-186 is redundant. Kindly delete them.
-In section 2.3.2, Kindly state the ethical issues that arose and the consent process of the participants.
-In the discussion section, replace the word participants with interlocutors.
-In line 658, what do you mean by "Many participants commented on the utilization of community networks....."? How many are many?
-Recommendations should come after the conclusion.
-A general concern with the title: where do you contextualize this study? Can readers easily know the context of this study from the title?
Thank you.
Reviewer
Author Response
Reviewer 2:
Comments and Suggestions for Authors
Thank you for this interesting article.
The article addresses an important health concern in maternal health. Through key informants as the participants, this article set-out to determine the feasibility, acceptability, and perceived benefits and risks of asking about Maternal mental health (MMH) as part of the routine assessment for infant malnutrition, especially in settings where specialist MMH services are unavailable, limited, or difficult to access.
Thank you authors, for taking the time to put this piece together. However, there are a few issues that you need to look into. Below are my specific comments.
Abstract- This section is fairly written, with the major components of a good abstract included. This section can pass in its present form.
Response
Thank you for this supportive and encouraging review. We really appreciate your time and feedback. We are especially glad that our abstract is good in its’ current form since this is the part of the article that many readers will focus on first / mainly to understand our key findings and key messages.
Other issues are addressed below – we agree that the edits help further strengthen our paper so thank you for the suggestions.
Introduction:
-Kindly insert a citation in line 34.
Response
References to our opening statement in lines 33,34 are already covered in detail by (original) references 1,2 and 3 in subsequent lines. However, following this helpful suggestion we have added a new reference to the opening line which highlights the need to better link mental health and nutrition in future policy/practice/ and public health programming.
-Remove data collection methods mentioned in the study design, i.e., semi-structured questionnaires, in-depth interviews etc. take them to data collection methods. Justifying why you chose a cross-sectional design and not others will also be essential.
Response
Thank you for this helpful suggestion. We have now
- added a line justifying our particular study design (last line of section 2.1). The focus is however on the fact that it’s a key informant study rather than the fact it’s cross-sectional. The vast majority of key informant studies are like ours, cross-sectional in nature (rather than longitudinal following the same key informants over time) so discussion of that would be more distracting than helpful.
- Also added a new reference to key informant studies – we trust this will help readers better understand our work as well as help design future studies of their own.
- Moved the statements pertaining to analysis strategy to the later “data analysis” subheading – they’re indeed a much better fit there rather than under “study design” subheading.
-Which sampling strategy led to 10 participants in this study? Additionally, describe how you conducted snowball sampling and its relevance.
Response
This is described in lines 159-160 and in the subsequent paragraph: the initial sampling was via authors’ professional networks of experts working in the field of nutrition/mental health. We have added an explanation of snowball sampling and a reference – again we trust that this will not only help current readers of our work but future researchers following up our work.
-Information in lines 181-186 is redundant. Kindly delete them.
Response
This has been done as suggested.
-In section 2.3.2, Kindly state the ethical issues that arose and the consent process of the participants.
Response
Prior to the interview participants read an ethics-committee-approved study information sheet and gave informed consent to participate. All the key informants were fellow health professionals and hence there was no power imbalance between researchers/participants compelling people to take part or answer any questions they were not fully comfortable with. Furthermore, we made it clear were only asking for professional opinions rather than personal experiences. Overall thus the study was very low risk from an ethical perspective.
We have added a few lines on this to the end of section 2.3.2 so that readers are clear about these important issues. Thanks for this suggestion.
-In the discussion section, replace the word participants with interlocutors.
Response
We are happy to do this if instructed by editors but would strongly prefer to keep our original phasing of “participants” so as to:
- be consistent with the rest of the text in methods and results
- be clearer to non-qualitative researchers: all researchers/readers of all methodological backgrounds know very well what a study “participant” is but only qualitative specialists will use the term “interlocutor”.
- keep the language as accessible as possible to as wide an audience as possible (e.g. as flagged in this guide, “interlocutors” is “a bit jargony” https://guides.tricolib.brynmawr.edu/c.php?g=877276&p=6300197
Since an edit here would not change the overall interpretation or messages arising from our paper, and with the above points in mind, we trust it is OK to reflect our own personal writing style here. Thank you for your understanding.
-In line 658, what do you mean by "Many participants commented on the utilization of community networks....."? How many are many?
Response
Some commented on this issue directly and some indirectly talked about this theme so it would be difficult to put an exact number on this. Since this is a qualitative hypothesis-generating study rather than a hypothesis-testing quantitative study the final message does not anyway differ whether 6/10 or 8/10 key informants talked about this.
-Recommendations should come after the conclusion.
Response
We are happy to do this if it better fits the journal style as instructed by editors. However, we believe that the current arrangement is preferable and more accessible to readers since:
- The recommendations section is in-depth and long and is aimed at readers wishing to explore the topic in detail. Hence it belongs better with the rest of the discussion.
- Many readers of academic literature only have time to focus on the final conclusions section. Hence we want to end our paper with a shorter section which better and more powerfully summarises the work overall and its implications. (please note also that we do give some high-level summary recommendations in the conclusions)
-A general concern with the title: where do you contextualize this study? Can readers easily know the context of this study from the title?
Response
This is a good point. The context is detailed in the abstract and in the keywords: between these, we trust that it is very clear where our work is set./where these issues apply. The title sadly is already at the word count limit and it would not be possible to expand it further.
Thank you.
Response
Many thanks to you too for taking time to read in detail and provide helpful comments on our work. It is better as a result of the changes suggested.
Reviewer 3 Report
Comments and Suggestions for Authors
Firstly, congratulations to authors on a job well done! This paper is a testament to the meticulous research and insightful analysis conducted by the authors. While the content is commendable, minor corrections are needed to address a few issues.
Multiple cited references should all be in one set of brackets, for example, instead of writing “[5],[6],[7],[8]” it should be [5-8].
Overall, the formatting of the text needs to be revised.
Lines 68-89: The text should be paraphrased so that the accented things from the guides are shortened, highlighting important things. It should be written as a regular text paragraph, making recommendations more understandable and concise for the reader, engaging for both experts and general readers alike.
Lines 289-290:”P6 stated that, whilst staff were happy to learn and adopt the new 289 skills, the implementation of these skills was inadequate.”- It would be useful to clarify in which way the implementation was inadequate, as it would bring light to other certain issues within this area.
Lines 309-310: “P9 also reported on the influence of religion and tradition, and its effect on delaying interventions from medical professionals.” – If possible, disclose the mentioned reasons for the delay of seeking medical attention.
Lines 331-332: Consider rephrasing the sentence for improved clarity, particularly in accordance to further text. It would be useful to specify which sentiment P5’s experience supports, weather about mental health becoming a priority or the challenges faced.
Headline “5. Conclusions” should be in bold.
Line 784-786: Conclusion section should only have authors’ conclusions, based on their research. Cited sentence at the end of the conclusion should be moved to the discussion section.
Lines 789-790: Given link for supplementary material is invalid, please correct this.
DOi numbers are missing from cited references, please adjust.
Comments on the Quality of English LanguageMinor editing of English may be required.
Author Response
Reviewer 3
Comments and Suggestions for Authors
Firstly, congratulations to authors on a job well done! This paper is a testament to the meticulous research and insightful analysis conducted by the authors. While the content is commendable, minor corrections are needed to address a few issues.
Response
We also thank reviewer 3 for this kind and generous support and appreciate the below suggestions to further improve our paper.
Multiple cited references should all be in one set of brackets, for example, instead of writing “[5],[6],[7],[8]” it should be [5-8].
Response
Thank you – this has been done throughout and indeed the paper looks neater with this format.
Overall, the formatting of the text needs to be revised.
Response
Thank you – with the edits we have made we trust all is now in order.
Lines 68-89: The text should be paraphrased so that the accented things from the guides are shortened, highlighting important things. It should be written as a regular text paragraph, making recommendations more understandable and concise for the reader, engaging for both experts and general readers alike.
Response
We are happy to cut or trim this section if instructed by editors but believe that the direct quotes as currently framed serve several useful and important purposes:
- They directly and powerfully show that mental health is in the new guidance (rather than readers having to take our word for it from a paraphrased section)
- They show that mental health is covered in multiple recommendations and is not an incidental one-off type mention.
- There is no potential for misunderstanding as might occur if the original guideline text is paraphrased.
- The direct quotes help boost knowledge and awareness of the key recommendations made by WHO. This is surely a good thing since the more readers know about these the better and the more likely that mental/malnutrition work will move forward in future.
Lines 289-290:”P6 stated that, whilst staff were happy to learn and adopt the new 289 skills, the implementation of these skills was inadequate.”- It would be useful to clarify in which way the implementation was inadequate, as it would bring light to other certain issues within this area.
Response
Thank you for highlighting this – we’ve included some more detail from the participant’s response, so hopefully this better explains that point.
Lines 309-310: “P9 also reported on the influence of religion and tradition, and its effect on delaying interventions from medical professionals.” – If possible, disclose the mentioned reasons for the delay of seeking medical attention.
Response
Excellent point - we’ve added a bit more context to this, and reworded it slightly to better explain why there is a delay in medical intervention.
Lines 331-332: Consider rephrasing the sentence for improved clarity, particularly in accordance to further text. It would be useful to specify which sentiment P5’s experience supports, weather about mental health becoming a priority or the challenges faced.
Response
Thank you. Hopefully our rewording of these sentences makes this point now clear.
Headline “5. Conclusions” should be in bold.
Response
This has been done and indeed helps highlight this key section
Line 784-786: Conclusion section should only have authors’ conclusions, based on their research. Cited sentence at the end of the conclusion should be moved to the discussion section.
Response
Thank you – we’ve reworded this sentence and moved the reference elsewhere for readers wanting to read more about the rights perspective to mental health (this paper was also published in Children jounral.)
Lines 789-790: Given link for supplementary material is invalid, please correct this.
Response
Thank you, we do have the interview guide available as a supplementary material but unfortunately had not had chance to submit this with our manuscript. This has been rectified now and the interview guide is available.
DOi numbers are missing from cited references, please adjust.
Response
We are using ENDNOTE reference manager which on our setup does not routinely present DOIs for all journal papers. We are happy to reformat if instructed by editors to fit an alternative journal-specific reference style.
Comments on the Quality of English Language
Minor editing of English may be required.
Response
We have done this as we addressed the comments raised by this and the other two reviewers. Thank you all again for your helpful suggestions.